

# Comparison between radiofrequency ablation and sublobar resections for the therapy of stage I non-small cell lung cancer: a meta-analysis

Shuang Chen[1], Shize Yang[2], Shun Xu[2] and Siyuan Dong[2]

[1] Department of Cardiology, The First Hospital of China Medical University, Shenyang, China
[2] Department of Thoracic Surgery, The First Hospital of China Medical University, Shenyang, China

## ABSTRACT

**Background**. Sublobar resection (SLR) and radiofrequency ablation (RFA) are the two minimally invasive procedures performed for treating stage I non-small cell lung cancer (NSCLC). This study aimed to compare SLR and RFA for the treatment of stage I NSCLC using the meta-analytical method.

**Methods**. We searched PubMed and Embase for articles published till December 2019 to evaluate the comparative studies and assess the survival and progression-free survival rates and postoperative complications (PROSPERO registration number: CRD42018087587). A meta-analysis was performed by combining the outcomes of the reported incidences of short-term morbidity and long-term mortality. The fixed or random effects model was utilized to calculate the pooled odds ratios (OR) and the 95% confidence intervals.

**Results**. Four retrospective studies were considered in the course of this study. The studies included a total of 309 participants; 154 were assigned to the SLR group, and 155 were assigned to the RFA group. Moreover, there were statistically significant differences between the one- and three-year survival rates and one- and three-year progression-free survival rates for the two groups, which were in favor of the SLR group. Among the post-surgical complications, pneumothorax and pleural effusion were more common for the SLR group, while cardiac abnormalities were prevalent in the RFA group. There was no difference in prevalence of hemoptysis between SLR and RFA groups, which might be attributed to the limited study sample size.

**Conclusion**. Considering the higher survival rates and disease control in the evaluated cases, surgical resection is the preferred treatment method for stage I NSCLC. RFA can be considered a valid alternative in patients not eligible for surgery and in high-risk patients as it is less invasive and requires shorter hospital stay.

# INTRODUCTION

Non-small cell lung cancer (NSCLC) continues to be the leading cause of death among all malignant neoplasms, thus focusing the attention of scientists and physicians

Corresponding author
Siyuan Dong, sydong@cmu.edu.cn

worldwide (*Torre, Siegel & Jemal, 2016*). Considering the rapid development of science and technology, various diagnostic techniques, including low-dose computed tomography(CT) and positron emission tomography-CT, have been used to detect early-stage pulmonary malignancies (*Menezes et al., 2010*). The cornerstone of NSCLC therapy is surgical resection, and the "gold standard" surgical approach for stage I NSCLC is lobectomy with systematic mediastinal lymphadenectomy as it offers the best chance of cure (*Howington et al., 2013*). However, resection is possible in only 20% of patients diagnosed with stage I NSCLC. Patients who are candidates for lobectomy are often elderly and have a history of atherosclerotic cardiovascular disease, pulmonary dysfunction, or comorbidities related with cigarette smoking. Thus, less invasive modalities are preferred for the treatment of these patients, such as sublobar resection (SLR) (*Harada et al., 2005*), radiofrequency ablation (RFA) (*Kodama et al., 2014*), and stereotactic body radiation therapy (*Chang et al., 2015*; *Grills et al., 2010*).

SLR, which is also referred to as 'limited resection,' is preferred for patients who cannot tolerate lobectomy with systematic mediastinal lymphadenectomy, as it preserves pulmonary function. SLR can be performed by anatomical segmentectomy or non-anatomical wedge resection, using open thoracotomy or applying video-assisted thoracoscopic surgery (VATS) approach. RFA is a minimally invasive approach using CT-guided percutaneous placement of an electrode into the lesion and generation of high-dose energy to cause coagulation necrosis (*Ambrogi et al., 2011*). Fernando's study (*Fernando et al., 2004*) presented RFA as the treatment choice for patients with stage I-II NSCLC, who are not eligible for surgery. In combination with other treatments, RFA can also be used to control a peripheral lesion in patients with advanced neoplasm (*Hiraki et al., 2014*).

For patients who are not eligible for lobectomy, SLR is recommended over non-surgical therapy (*Donington et al., 2012*). However, the superiority of SLR over RFA is still controversial (*Kim et al., 2012*; *Zemlyak, Moore & Bilfinger, 2010*). Presently, there is no evidence from randomized controlled trials (RCTs) proving the RFA results better than SLR results in treating stage I NSCLC. This study aimed to evaluate the postoperative complications and survival rates of patients with stage I NSCLC, who underwent SLR or RFA.

## MATERIALS & METHODS

### Information sources and search strategy

Our research was registered on PROSPERO (CRD42018087587). Electronic searches were conducted by two investigators (Shuang Chen and Sheze Yang) using PubMed, Cochrane Library, Web of Science, Ovid MEDLINE, Google Scholar, and Embase databases until December 2019. The Medical Subject Heading terms and keywords included in our search strategy in a variety of combinations were "radiofrequency ablation," "sublobar resections," "non-small cell lung cancer," "stage I," "wedge resection," "segmentectomy," "RFA," and "SLR." All related scientific original articles, reviews, animal studies, letters, opinion pieces, and editorials were researched. Only manuscripts written in English were

considered. Moreover, the Science Citation Index was used to further cross-reference the studies which met the pre-defined inclusion criteria.

## Study selection

Randomized intervention studies and observational cohort studies were eligible for inclusion if they followed-up patients for at least 3 years. Thus, the pre-defined inclusion criteria established for our research were as follows:

(1) Studies that compared the survival situation and postoperative complication rates of SLR and RFA in patients with stage I NSCLC;

(2) Articles documenting survival data from reports presented at major radiology and thoracic surgery academic conferences (RSNA, AATS, and EACTS) or studies that were published in peer-reviewed publications;

(3) Studies comprising participants with similar main clinical characteristics and without a history of malignant tumors;

(4) Studies that reported at least one of the following results: survival, progression-free survival, and local recurrence rates (full texts were retrieved from studies that met all the inclusion criteria; SLR refers to wedge resection and segmentectomy);

The exclusion criteria were:

(1) Studies not comparing SLR or RFA as intervention methods and studies evaluating stages II, III, and IV NSCLC;

(2) Research focusing on participants' treatment for pulmonary metastases;

(3) Articles with overlapping information on researchers, hospitals, institutions, or participants' cohorts (only the most informative and newest literature was considered);

(4) Articles published in the past 20 years regarding recent remarkable technological advancements.

## Data extraction and quality assessment

All the related studies were searched and reviewed by two independent investigators (Shuang Chen and Shize Yang) according to our pre-defined eligibility criteria. The investigators also extracted research data, study design, and baseline characteristics (age and sex) and endpoints according to the predesigned data extraction form. Any needed but incomplete survival information in the articles was acquired by directly contacting the author. Differences in data extraction results between the two investigators were solved through discussion between them and finally overseen by a third senior independent author (Siyuan Dong). The final extracted data were confirmed by two senior investigators (Siyuan Dong and Shun Xu). The selected studies were assessed for quality by applying the Downs and Black quality assessment method we had used in our previous studies (*Dong et al., 2015*; *Dong et al., 2014*; *Downs & Black, 1998*). The progression-free survival was defined as the period from the date of the initial surgical resection or RFA until the date of recurrence.

## Data synthesis and statistical analysis

Review Manager version 5.3 software package (Cochrane Collaboration Software) was utilized in this study. Survival data were reported as hazard ratios and dichotomous

clinical outcomes as risk ratios. The corresponding 95% confidence intervals (95% CI) were calculated. Moreover, a $p$ value <0.05 was considered statistically significant when assessing the value between SLR and RFA. A fixed-effects model was adapted if there was no statistically significant difference in terms of heterogeneity ($p > 0.05$). Otherwise, a random-effect model was adopted. Heterogeneity between all the included articles was investigated using the $I^2$ statistic with statistical significance $P < 0.05$. Hence, the upper thresholds for low, moderate, and high heterogeneity were defined as $I^2$ values between 25% and 50%, between 50% and 75%, and greater than 75%, respectively.

## Publication bias
Visual inspection of funnel plots was applied to assess potential publication bias.

# RESULTS
## Study characteristics
Four retrospective cohort studies that met our inclusion criteria were included in our study between 2010 and 2015: two from the USA and one each from Germany and Italy. A total of 309 participants were included in the research: 154 were assigned to the SLR group, and 155 were assigned to the RFA group to assess the postoperative complications and survival rates. As suggested by the Preferred Reporting Items for Systematic Reviews and Meta-Analyses statement (*Liberati et al., 2009*), we presented the process of identification and inclusion of studies in a flow diagram (Fig. 1). The articles' evaluation index and basic characteristics and are presented in Table 1. Alexander's (*Alexander et al., 2013*), Ambrogi's (*Ambrogi et al., 2015*), and Safi's (*Safi et al., 2015*) research were in favor of SLR; however, in Zemlyak's (*Zemlyak, Moore & Bilfinger, 2010*) study, RFA demonstrated a comparable effect to that of SLR in participants with stage I NSCLC.

## Assessment of complications
These four studies compared the postoperative complications between the two groups. The postoperative complications included pneumothorax, hemoptysis, pleural effusion, and postoperative cardiac abnormality. Two papers (*Alexander et al., 2013*; *Zemlyak, Moore & Bilfinger, 2010*) documented the results of hemoptysis (OR = 0.32; 95% CI [0.03–3.17]; $p$ = 0.33) with some heterogeneity ($x^2 = 0.39$, $p = 0.53$, $I^2 = 0\%$, Fig. 2A). All four papers documented the results of pneumothorax (OR = 0.14; 95% CI [0.06–0.30]; $p < 0.00001$) with some heterogeneity ($x^2 = 7.01$, $p = 0.07$, $I^2 = 57\%$, Fig. 2B). Three papers (*Alexander et al., 2013*; *Ambrogi et al., 2015*; *Safi et al., 2015*) documented the results of pleural effusion (OR = 0.24; 95% CI [0.06–0.98]; $p = 0.05$), with some heterogeneity ($x^2 = 0.46$, $p = 0.80$, $I^2 = 0\%$, Fig. 2C). Three papers (*Alexander et al., 2013*; *Ambrogi et al., 2015*; *Safi et al., 2015*) documented the results of cardiac abnormality (OR = 13.09; 95% CI [2.45–69.94]; $p = 0.003$) with some heterogeneity ($x^2 = 0.86$, $p = 0.65$, $I^2 = 0\%$, Fig. 2D). Only postoperative cardiac abnormality was prevalent in the RFA group, and pneumothorax and pleural effusion were both prevalent in the SLR group. Hemoptysis incidence was the same for both groups.
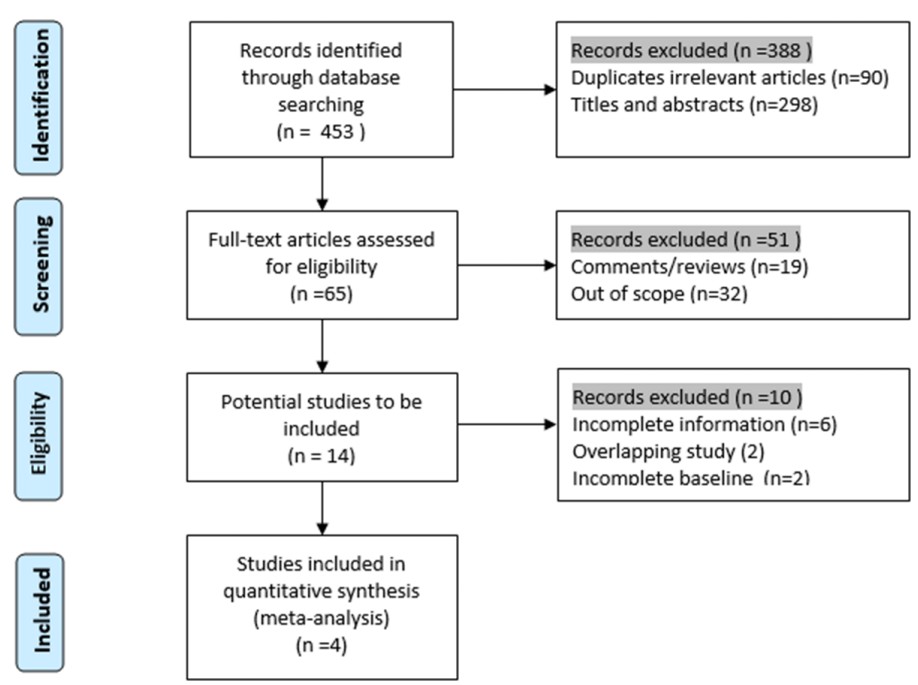

**Figure 1** Flow chart of the identification of researches for inclusion.

**Table 1** The evaluation index and characteristics of the included researches.

| Study | Design | Country | NO | Gender (M/F) | Mean age (years) | Tumor sizer (mm) | FEV1/ Predicted (%) | Hospital stay (Day) | Assessment score |
|---|---|---|---|---|---|---|---|---|---|
| *Zemlyak, Moore & Bilfinger (2010)* | OC | USA | S 25 | S 9/16 | S 66.0 | NR | S 65 | 6 | 14 |
| | | | R 12 | R 7/5 | R 74.0 | NR | R 64 | 1.8 | |
| *Alexander et al. (2013)* | OC | USA | S 28 | S 12/16 | S 73.8 | NR | S 54 | 5 | 18 |
| | | | R 56 | R 24/32 | R 77.6 | NR | R 52 | 0 | |
| *Ambrogi et al. (2015)* | OC | Italy | S 59 | S 46/13 | S 70.0 | 26 | S 47 | 6 | 19 |
| | | | R 62 | R 45/17 | R 76.0 | 23 | R 49 | 2 | |
| *Safi et al. (2015)* | OC | Germany | S 42 | S 27/15 | S 69.6 | 19 | S 69 | NR | 15 |
| | | | R 25 | R 34/15 | R 71.2 | 22 | R 67 | NR | |

**Notes.**
S, SLR; R, RFA; M, Male; F, Female; OC, Observational cohort; NR, Not reported.

## Assessment of survival and recurrence

All four studies reported the results of the 1-year survival rate (*Alexander et al., 2013*; *Ambrogi et al., 2015*; *Safi et al., 2015*; *Zemlyak, Moore & Bilfinger, 2010*)), and no significant heterogeneity was observed among them ($x^2 = 2.15$, $p = 0.54$, $I^2 = 0\%$). Thus, a fixed-effects model was adopted (OR = 3.34; 95% CI [1.13–9.89]; $p = 0.03$, Fig. 3A). These four studies also reported the outcomes of the 3-year survival rate, and heterogeneity was calculated through the research ($x^2 = 4.18$, $p = 0.24$, $I^2 = 28\%$), using the fixed-effects model (OR = 1.95; 95% CI [1.20–3.18]; $p = 0.007$, Fig. 3B). Only one paper

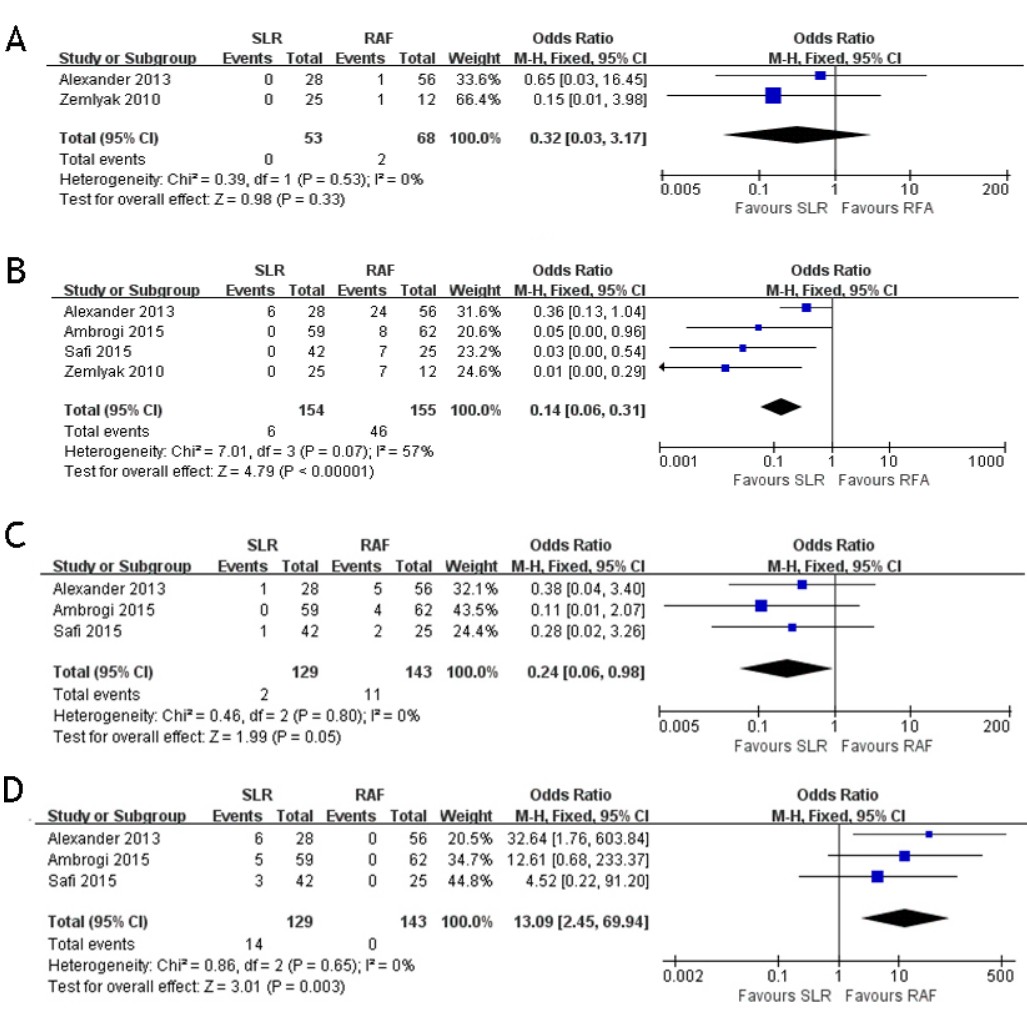

**Figure 2** Hemoptysis (A), pneumothorax (B), pleural effusion (C) and cardiac abnormality (D) forest plot of the Odds Ratio (OR) following SLR versus RFA for stage I NSCLC.

(*Ambrogi et al., 2015*) documented the results of the 5-year survival rate(52% for SLR and 35% for RFA); hence, the results could not be consolidated. All the results showed greater prevalence in the SLR group.

Three articles (*Ambrogi et al., 2015*; *Safi et al., 2015*; *Zemlyak, Moore & Bilfinger, 2010*) compared the 1-year progression-free survival rate (OR = 2.72; 95% CI [1.18–6.29]; $p = 0.02$), and no significant heterogeneity among these articles was detected ($x^2 = 0.34$, $p = 0.84$, $I^2 = 0\%$, Fig. 4A). All three studies also presented the 3-year progression-free survival rate (OR = 3.01; 95% CI [1.63–5.55]; $p = 0.0004$); however, there was no significant heterogeneity among subjects treated with SLR and those treated with RFA ($x^2 = 2.00$, $p = 0.37$, $I^2 = 0\%$, Fig. 4B). Nevertheless, significant 1- and 3-year progression-free survival rate benefits were observed in the SLR group. We also intended to compare the 5-year progression-free survival rate results between the two groups. However, only one article

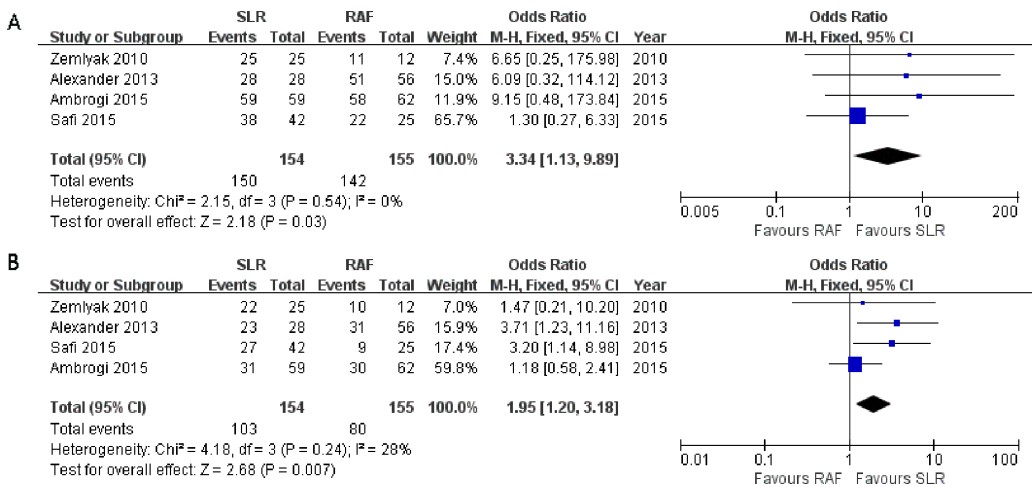

**Figure 3** One- (A) and three-year (B) survival rate Forest plot of the Odds Ratio (OR) following SLR versus RFA for stage I NSCLC.

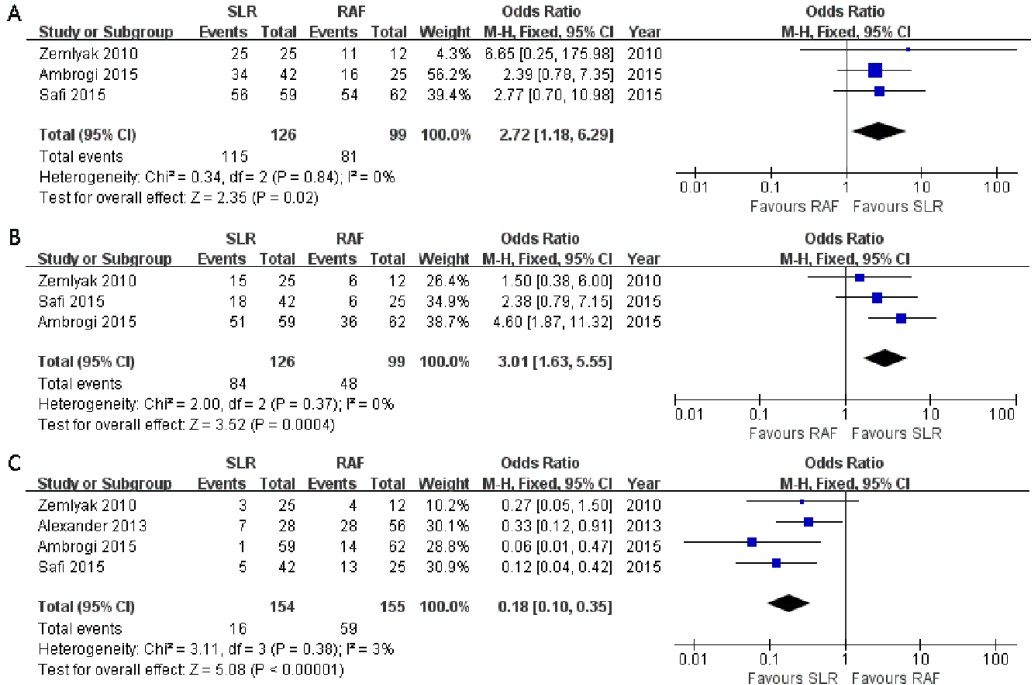

**Figure 4** One- (A) and three-year (B) progression-free survival rate and local recurrence (C) forest plot of the Odds Ratio (OR) following SLR versus RFA for stage I NSCLC.

documented it, and it was in favor of SLR (*Ambrogi et al., 2015*). All the survival rate results are presented in Table 2.

Considering that local recurrence is one of the most important aspects for evaluating the NSCLC treatment efficacy, it was assessed in all four studies. There was no significant

**Table 2  Summary of the outcomes between SLR and RFA of patients with stage.**

| Variables | Results | | OR | P-value | $I^2$ |
|---|---|---|---|---|---|
| | SLR | RFA | | | |
| 1-y survival | 97% | 91% | 3.34 | 0.03 | 0% |
| 3-y survival | 67% | 52% | 1.95 | 0.007 | 28% |
| 1-y progression-free survival | 91% | 81% | 2.72 | 0.02 | 0% |
| 3- y progression-free survival | 67% | 48% | 3.01 | 0.0004 | 0% |

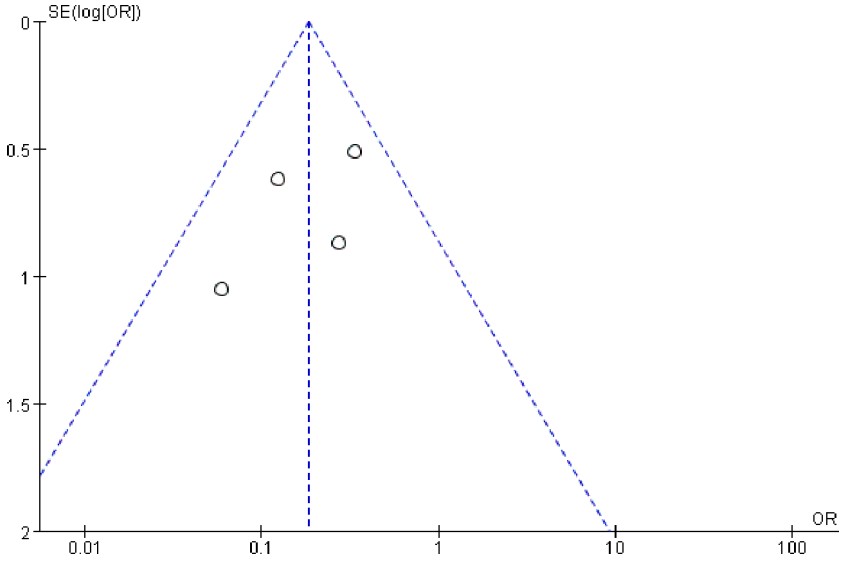

**Figure 5  Funnel plot of the outcome of one-year survival rate.**

heterogeneity between these four studies ($x^2 = 3.11$, $p = 0.38$, $I^2 = 3\%$), and the fixed-effects model was adopted. The combined result was in favor of SLR (OR = 0.18; 95% CI [0.10–0.35]; $p < 0.00001$, Fig. 4C). We tried to consolidate the distant recurrence, but only one article in favor of SLR had documented it (*Zemlyak, Moore & Bilfinger, 2010*). Hence, the available data was not suitable for further research.

## Publication bias

Publication bias is possibly observed when non-significant outcomes remain unpublished, inevitably amplifying the evident magnitude of a function. This funnel plots of this study are demonstrated in Fig. 5. The funnel plots of the 1-year survival rate following SLR and RFA for the treatment of stage I NSCLC manifested a slight asymmetry, and all points were within 95% CI, suggesting a low publication bias.

## DISCUSSION

With the development of low-dose CT technology, early stage NSCLC can be immediately detected and diagnosed (*Veronesi et al., 2008*). Ever since the widely quoted research from the Lung Cancer Study Group was published in 1995, the standard treatment for patients

with NSCLC has been lobectomy (*Ginsberg & Rubinstein, 1995*); however, some patients are at a high risk and refuse to undergo radical resection. Considering that only 20% of NSCLC patients are eligible for lobectomy, owing to complex clinical histories, some less invasive approaches are currently being developed, including RFA and SLR. These two approaches have shown promising results (*Jones et al., 2015*; *Speicher et al., 2016*).

RFA is a percutaneous treatment performed with local anesthesia and conscious sedation (*Choe et al., 2009*; *Palussiere et al., 2015*). Those highly in favor of RFA emphasize its definitive advantages over surgery, including outpatient treatment and the percutaneous performance of this procedure, using local anesthesia and avoiding thoracotomy for patients who refuse surgical resection or present severe comorbidities. RFA does not significantly affect the patient's cardio-pulmonary function, and it is also associated with a significant decrease in the length of hospital stay. Moreover, RFA allows the ablation of lesions without major damage to the peripheral normal tissues (*Simon et al., 2007*; *Wan et al., 2016*; *Wan, Wu & Zhang, 2016*). RFA complications are relatively minor and acute, although their frequency is considerable. Zemlyak's study (*Zemlyak, Moore & Bilfinger, 2010*) showed that most patients undergoing RFA could be discharged within 24 h of therapy, and RFA has similar overall and cancer-specific survival rates to those of SLR. Considering that RFA does not lead to any loss of pulmonary function and that it can be repeatedly performed, if a patient has a tumor recurrence or a new tumor growth, it also has the following advantages: it is well tolerated by outpatients and is complementary to chemotherapy, used to treat metachronous and synchronous lesions, results in rapid recovery of physical performance, and has a relatively short treatment time (*Cheng, Fay & Steinke, 2016*; *Chua et al., 2010*; *Li et al., 2013*; *Ridge et al., 2014*). The four complications documented in the articles we included were hemoptysis, pneumothorax, pleural effusion, and cardiac abnormality. The most common complication was pneumothorax: it was observed in every article and more notably in the RFA group than in the SLR group (Fig. 2). Although the RFA group had more complications than the SLR group, they were relatively minor.

However, the following disadvantage is notably observed in RFA: the application of RFA is limited considering the proximity of a vascular structure to the location and size of the tumor. The energy generated to the tumor will be reduced if there is a vessel with diameter greater than 0.3 cm, owing to the loss of energy through convection within the surrounding circulatory system (*Lencioni et al., 2004*). The proximity of the tumor to the trachea, heart, and esophagus increases the risk when performing RFA. Heat is also reduced if the tumor is greater than three cm in diameter considering its periphery; hence, it is difficult to reach an ideal ablative temperature, resulting in an impaired local control and diminished response (*Sharma, Abtin & Shepard, 2012*). The advantage of SLR is that it has better oncologic outcomes than RFA. According to Keenan's research (*Keenan et al., 2004*) comprising 201 stage I NSCLC patients, the advantages of SLR are that it better preserves lung function than does lobectomy and the forced vital capacity and forced expiratory volume in 1 s are better preserved in the SLR group than those in the RFA group. Moreover, Kodama's (*Kodama et al., 1997*) and Kates's (*Kates, Swanson & Wisnivesky, 2011*) studies both reveal that for T1N0 NSCLC, the survival rate of SLR is also similar to that of lobectomy.

Recently, several authors reported different VATS techniques to localize lung tumors, avoiding the necessity of open thoracotomy, which can be performed through minimally invasive video-assisted thoracoscopic surgery (*Ambrogi et al., 2012*; *Gruber-Rouh et al., 2017*; *Lin et al., 2016*; *Refai et al., 2020*). Furthermore, an incidence rate of 5% of lymph node involvement was observed in 100 NSCLC patients with tumor <1 cm in diameter, suggesting that nodal assessment should be taken into consideration even in small lesions (*El-Sherif et al., 2005*). Hence, an additional advantage of SLR is that lymph node sampling can be performed at the time of surgical resection, allowing the identification of potential metastatic nodes and more precise staging to guide treatment. The main disadvantage of surgery is that not every patient can tolerate resection secondary to comorbid disease or poor reserve. Additionally, the SLR significantly requires a longer post-procedure length of stay and higher cost than RFA (*Alexander et al., 2013*).

Kim believed that RFA has a survival rate similar to that of surgical treatment in stage I NSCLC patients, specifically in those not eligible for surgical treatment (*Kim et al., 2012*). To confirm this hypothesis, all the clinical stage II–III NSCLC were excluded, considering that surgery is the standard treatment for these patients. Hence, our study differs from previous meta-analyses and studies, as all the included patients had stage I NSCLC, which is undoubtedly considered as early-stage NSCLC. We also considered other factors that influence patient outcomes, such as age, tumor size, and clinical condition. The SFR group patients are significantly older than the SLR group patients; however, the other baseline characteristics are similar between the two groups (Table 1). Moreover, according to our study, higher local recurrence is observed in the RFA group than in the SLR group, and longer survival and progression-free intervals are more frequently observed in the SLR group than in the RFA group. The possible reason for these hypotheses might be the existing inherent selection bias related to the neoplasm operability being defined by surgical intervention, which results in a decision algorithm stating that RFA is only to be performed after ruling out surgery. This possible reason was confirmed by a study conducted in Massachusetts General Hospital (*Lanuti et al., 2009*) in 2009, reporting of a group of 31 patients treated with RFA who were deemed "not eligible" for surgery. In this study, only three patients died of disseminated lung cancer. In all these four studies, the age of the participants in the RFA group was greater than that of the participants in the SLR group (Table 1), supporting the hypothesis that elderly patients are at higher risk of death, due to comorbidities, rather than lung cancer, as reflected in the RFA group in Alexander's study (*Alexander et al., 2013*). Moreover, the assessment of stages was more accurate for the SLR group than for the RFA group. Thus, the RFA group may be understated, resulting in survival bias. Finally, incomplete tumor ablation may also affect the survival rate.

This is the first meta-analysis that compares SLR and RFA for stage I NSCLC. We assessed the clinical results of patients with stage I NSCLC. We compared the survival rates of SLR with those of CT-guided thermal ablation in four studies. RFA, despite its higher disease recurrence and lower survival rate, can be considered a valid alternative for inoperable and high-risk patients due to its short hospital stay and low invasiveness.

Despite the outcomes of our study, there are some limitations. First, RCTs comparing RFA and SLR have not been conducted. Most available studies are single-institution

case series and small observational studies. Second, due to the retrospective design of the included studies, patients undergoing RFA were older and had higher co-morbidity scores and lower performance than patients undergoing SLR; hence, those undergoing RFA were categorically likely to die sooner considering that they were not eligible for surgery. Although all the included studies focused on stage I NSCLC, the different stages of NSCLC also influence the patients' survival and local recurrence. Third, the included studies exclusively focused on stage I NSCLC. It is known that the therapeutic effect of RFA is also closely associated with tumor size, but studies assessing the association between RFA and tumor size have not been conducted to date. Finally, the mediastinal and hilar lymph nodes are evaluated by preoperative imaging and not by pathologic verification when RFA is performed. Hence, the real staging status of the patients may be underestimated, resulting in survival bias. Moreover, with RFA, the tumor is not eliminated, and a residual scar, which is commonly noted after treatment, can be mistaken for a recrudescent lesion. The inclusion of additional RCTs to the studies we evaluated would have increased the significance of our results.

## CONCLUSION

The outcomes of our study affirm that surgical intervention, even if just limited to SLR, has a better survival rate than RFA; therefore, it should be the preferred choice of treatment in patients, excluding those not eligible for surgery. Future large-scale prospective randomized studies would help compare the survival rates of the different approaches and define better the participants considered in the high risk group.

### Funding

This article was supported by the National Natural Science Foundation of China (81702242). The funders had no role in study design, data collection and analysis, decision to publish, or preparation of the manuscript.

### Grant Disclosures

The following grant information was disclosed by the authors:
National Natural Science Foundation of China: 81702242.

### Competing Interests

The authors declare there are no competing interests.

### Author Contributions

- Shuang Chen and Shize Yang performed the experiments, analyzed the data, prepared figures and/or tables, authored or reviewed drafts of the paper, and approved the final draft.
- Shun Xu conceived and designed the experiments, prepared figures and/or tables, and approved the final draft.

- Siyuan Dong conceived and designed the experiments, authored or reviewed drafts of the paper, and approved the final draft.

## Data Availability

The raw data is available in the Results section of the article and in Figs. 1–5.

## Supplemental Information

Supplemental information for this article can be found online at http://dx.doi.org/10.7717/peerj.9228#supplemental-information.

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
