# Peer review of "Comparison between radiofrequency ablation and sublobar resections for the therapy of stage I non-small cell lung cancer: a meta-analysis"

_PeerJ, doi:10.7717/peerj.9228_

## Round 0.1 · original submission · Major Revisions

The reviewers have recommended publication pending major revisions. Therefore, I invite you to respond to the reviewers' comments at the bottom of this letter and revise your manuscript accordingly.

Reviewer 1 ·

Basic reporting

Major language editing is necessary.

ABSTRACT
line 12 This research aims to compare SLR and RFA for treatment of stage I NSCLC, applying the...
line 32 And There were significant statistical differences... (one-, three- and five-year)... (one- and three-year). [three-year or three years survival]
line 34 the postoperative complications (hemoptysis, ...) are in foavour of SLR...

INTRODUCTION
line 55 scientists and physician (or medical doctor)

line 57 Though, surgical treatment provides patients with the best chance of total cure and long-term survival. This sentence is not clear. I suggest to cancel it, adding "offering the best chance of cure" at the end of the second sentence.

The cornerstone of therapy in NSCLC patients is surgical resection and the “gold standard” surgical approach for stage I NSCLC is lobectomy with systematic mediastinal lymphadenectomy, offering the best chance of cure

line 63 are often old (not:predisposed to advanced age), and affected by atherosclerotic cardiovascular disease, pulmonary dysfunction and other comorbidities related with cigarette smoking.

Line 86 Sub-lobar resection (SLR), also known to as limited resection, is preferred for
69 patients that cannot tolerate lobectomy with systematic mediastinal
70 lymphadenectomy specifically to preserve pulmonary function. SLR can be
71 conducted by anatomical segmentectomy or non-anatomical wedge resection, through an open thoracotomy or with a VATS approach

Line 73 placement of an electrode into the lesion and generation of high-dose energy to cause
74(not: tumor to undergo) coagulation necrosis (not:and cell death)[
Fernando’s[9] study
75 showed that RFA is a choice for the patients with 76 stage I-II non-small cell lung cancer, who are not suitable for operation. RFA can also be used for local control of
77 peripheral lesion in patients with more advanced stage in combination with
78 other treatment.

NOT: With the aim of exploring whether RFA can get the survival comparable to 84 that by SLR to stage I non-small cell lung cancer patients,
YES: we performed this research,
85 based on patients with stage I non-small lung cancer who underwent SLR or RFA to
86 evaluate both the postoperative complication and survival rates.

MATHERIALS AND METHOD
line 103 (1) Comparison of the survival situation and post-operative complication rates of SLR and RFA in patients with stage I NSCLC
104

105 (2) Articles that contain the survival data of the reports that presentations presented (?) at major
106 radiology and thoracic surgery academic conference (RSNA, AATS and EACTS) or
107 study published in peer-reviewed publications;

126 standard. the (you have already said that they are two) investigators also

RESULTS
I suggest to describe firstly Study characteristics, secondly Assessment of complication, and then assessment of survival.
I suggest to use this plot to develope the DISCUSSION

DISCUSSION

The application of less invasive treatment for
217 NSCLC is especially important given that only 20% of NSCLC are suitable for
218 lobectomy and some less invasive approaches are currently being developed for
219 patients with early-stage NSCLC whose basic conditions preclude them from being
220 cured in accordance with the standard of care.

I suggest to semplify the above mentioned sentence.: considering that only 20% of NSCLC patients are suitable for lobectomy due to poor clinical conditions, some less invasive approaches are currently being developed: RFA and SLR
NOT: The choice of a modality relays more on the 222 theoretical knowledge and personal custom of the doctor than on the proof. In the next page you say that many authors reserved RFA for patients that are deemed medically inoperable. I don't think that the treatment choise depends on personal custom of physiscians. I shouldn''t be.
.
I suggest to briefly describe the advantages and disadvantages of each procedure


...RFA is a percutaneous treatment performed with local anaestesia and conscious sedation

NOT (you have already said that): RFA induces heat denaturation of cellular proteins causeing tumor cell death, which 224 is used to provide palliation and could achieve excellent local control for patients
225 with inoperable NSCLC[24, 25].Those who advocate RFA stress it present some 226 definitive advantages over surgery.

RFA can be conducted percutaneously, avoiding 227 a thoracotomy for those who refuse surgical resection or with severe co-morbidities.
The RFA has less influence on the cardio-pulmonary function and lower cardiac abnormality rate. it’s also
229 associated with a significant decrease in length of stay
NOT: our results also reveal the 230 RFA group have a lower cardiac abnormality rate

Another feature of RFA
231 is that it allows ablation of lesions without major damage to peripheral normal 232 tissues[26].
The complications of RFA were relatively minor.
I suggest to briefly descibe the most frequent complications and the mean post-operative stay reported in the 5 reports you analysed, adding FIGURE 4-citation here
(I suggest to compare survival rates of the 5 reports, later)

This topic is unnecessary, remove it: Hu and associates initial explored the mechanism of RFA for 242 the treatment of primary pulmonary malignancy and lung metastasis from liver 243 cancer: upregulating tumor suppressor miRNAs (let-7a and miR-34a) and
244 downregulating onco-miRNAs (miR-21)[31].

...SLRs require a general anaeshesia. (this fact can be a disadvantage because many patients can tolerate a general anaesthesia) They can be performed through a thoracotomy or with a Video assisted thoracoscopic surgery (recently several authors reported different VATS techiniques to localise lung toumors avoiding the necessity of open thoracotomy [Surg Endosc. 2012 Apr;26(4):914-9. doi: 10.1007/s00464-011-1967-8 ] [J Thorac Cardiovasc Surg. 2016 Aug;152(2):535-544.e2. doi: 10.1016/j.jtcvs.2016.04.052 ] [Clin Radiol. 2017 Oct;72(10):898.e7-898.e11. doi: 10.1016/j.crad.2017.05.015 ][Radiol Med, 125 (1), 24-30  Jan 2020 DOI: 10.1007/s11547-019-01077-x]
SLR is associated with higher complication rate in respect to RFA..


after discussing about complications, I suggest to analyse survival and disease-free interval rates of RFA and SLR, trying to explain the reasons (example: 1 different stage clinical for RFA and pathologiacl for SLR; 2 incomplete toumor ablation for RFA; 3 patients medical conditions ..)


This part is excessive and redundant. I suggest to simplify it underlining that RFA can not assure the complete toumor ablation, while surgival resection can provide limph node staging.

REMOVE THIS: Those who are against the RFA stress that SLR could provide better oncologic 246 outcomes in respect to RFA. One of the key deficiencies for RFA is the difficulty in 247 achieving thoroughly lesion destruction[32]. As the expected temperature for 248 treatment cannot be completely achieved throughout the lesion due to periphery of 249 the lesion is far away from the core of ablation. On the contrary, insufficient RFA 250 can induce the growth of non-small cell lung
ancer via up-regulating HIF-1α 251 through the PI3K/AKT signals[33].

Experimental design

the authors clearly descibe the aim of the study . the issue is relevant and interesting

the literature review apperared performed in a rigorous way, with high ethical standard

Validity of the findings

interesting finding, supported by the results of the study

Additional comments

major language editing is necessary

Reviewer 2 ·

Basic reporting

1. The English language should be improved throughout the manuscript to ensure that an international audience can clearly understand your text. Grammatical mistakes such as preposition usage (Line20: “is aimed at assessing” not “is aimed to assessing”), misuse of pronoun reference (Line62, 297: “patients who are” not “patients which are”), confusing uses of articles (Line71-72: “RFA is a minimally” not “The RFA is a minimally”), as well as singular and plural errors (Line192: “Assessment of complications” not “Assessment of complication”) should all be rectified.
2. References should be added to Table 1, Line76-78, Line195-204, and Line 269.
3. Line267-269: The language should be improved as the current phrasing makes comprehension difficult.
4. Line 269-274: I suggest that you move these analyses of bias to the “limitations” part also in the “Discussion” section.
5. Line 291: “randomized clinical trials (RCTs)” not “RTC”
6. Figure 1 showing the process of study selection needs more detail. I suggest you provide reasons together with the number of articles in the exclusion process.
7. Table 1 presented with basic characteristics of patients’ needs improvement. I suggest that you list the characteristics of each group of patients in single rows and remember to provide a consistent display of numbers among different studies (For example: "median ± SD" for Age). Descriptive data such as tumor sizes, indicators of patients' clinical condition (ECOG, FEV1 or FEV1%), and numbers of patients who have received adjuvant treatments could also be added to the table.

Experimental design

1. Your most important issue is that the extent of operation for patients in the surgery group was “lobectomy or pneumonectomy” in one of your included studies (by Kim et al) which did not meet the including criteria. You have to exclude this study from your work.
2. Line 104: I suggest that you should mention the system you chose to identify the stage of included patients.
3. Line110-111: For the selection of clinical outcomes included in your study, I have the following suggestions. First, I suggest that both disease-free survival (DFS) and progression-free survival should be included since RFA probably cannot eliminate tumors. Second, "distant metastasis" may be considered together with the "local recurrence" when evaluating the efficacy of treatments, both of which were involved in the study by Ambrogi et al. The number of patients you extracted for meta-analysis was also the sum of the two cases. Please reconfirm the data included in the meta-analysis.
4. Line134-136: Disease-free survival seems to be a concept used to describe the period after a successful treatment during which there are no signs and symptoms of the disease that was treated.
5. Line166-191: As far as I know, most of the included studies in your work provided with survival rates or survival curves. Some studies even did not present with specific survival data of the year that you have chosen in your meta-analysis. Please explain in the “Methods” section how you extracted the number of patients of the year of one, three, and five for analysis. Besides, please reconfirm the accuracy of the data in your meta-analyses.
6. Line 174-175: Heterogeneity was classified into grades with low, medium and high in your study. You should better use these terms instead of "some".

Validity of the findings

The studies included by the authors on this issue were comprehensive but less representative. One of the studies included by the authors is not suitable for the study. Some of the data extracted by the authors for analysis seemed to be less accurate. Both of the two reasons result in the final results of this meta-analyses to be less robust or reliable.

Additional comments

The authors focused on comparisons on efficacy between RFA and sub-lobar resection in stage I NSCLC patients. The main objective of this manuscript has been straightforwardly achieved. This Reviewer believes that this manuscript could be of relevance to clinicians in this field. Nevertheless, there are still several major points to be justified and some minor points to be corrected.

Reviewer 3 ·

Basic reporting

no comment

Experimental design

no comment

Validity of the findings

no comment

Additional comments

This meta-analysis has performed systematic review of literature which compared the prognosis of RFA and that of SLR in NCLC patients in stage I. This study has clinical value and the results are promising. However, there are still some issues should be addressed.
1. Besides the treatment methods, what about the other risk factors influencing the outcome of patients with NCLC? What about the proportions of these risk factors in patients of your included researches?
2. It’s all known that the therapeutic effect of RFA is closely related to tumor size, so what is the tumor size of the tumors included in the study? Was there any difference between the two groups? These should be analyzed and discussed.
3. Minor suggestion for the “Abstract” section, “In cases of stage I NSCLC, surgical resection is preferred due to its higher survival rates and disease control as opposed to radiofrequency ablation. The RFA option could be considered a valid alternative for inoperable and high-risk patients, due to its short hospital stay and low invasiveness” Should be the content of “conclusions”, but not “results”.
4. When discussing the mechanism of RFA, the authors described that it’s “the upregulating rumor suppressor miRNAs and downregulating onco-miRNAs (Lines 241-244). However, we all known that the main mechanism of RFA for treating tumors should be thermally induced tumor cell death, and the molecular mechanism mentioned above should occur in areas where the temperature is not high enough to kill cells. Therefore, this argument tends to confuse readers.
5. Some grammar mistakes need serious correction, for example, “There are two different minimally invasive approaches, sub-lobar resection (SLR) and radiofrequency ablation (RFA), which are performed for stage I non-small cell lung cancer treatment” (Lines 18-20).
6. Full text should be checked and corrected for spelling errors, for example, “RAF” (Line 226).
7. In Lines 417-418, the figure legend of Figure 3 should be “…disease free survival rate…”, but not “…disease survival rate…”.

---

## Round 0.2 · Minor Revisions

The reviewers have recommended publication pending minor revisions. Therefore, I invite you to respond to the reviewers' comments at the bottom of this letter and revise your manuscript accordingly.

Reviewer 2 ·

Basic reporting

The article was well-written, but there are still some minor points to be improved:
1. In lines 97 and 189, it was written "adapted". I think they were typos and the intention was to write "adopted".
2. There're some grammar errors in the manuscript, so the English writing should be polished further.

Experimental design

The research question was novel and well defined. Analyses were performed with a high technical and ethical standard.

Validity of the findings

The results were presented in a convincing and rigorous manner. Limitations of this work were also fully discussed by the authors.

Additional comments

This work investigated the optimal choice from RFA and sub-lobar resection in stage I NSCLC patients, which would be an interesting topic for thoracic surgeons. The article was well written, and the results of the meta-analyses were presented in a convincing and rigorous manner. I have only minor comments to make.

---

## Round 0.3 · accepted · Accept

The statistical analyses in this manuscript are OK. The English writing is greatly improved in this version.